# SCIURus: Shared Circuits for Interpretable Uncertainty Representations in Language Models

## Abstract

We investigate the mechanistic sources of uncertainty in large language models (LLMs), an area with important implications for their reliability and trustworthiness. To do so, we conduct a series of experiments designed to identify whether the factuality of generated responses and a model's uncertainty originate in separate or shared circuits [5] in the model architecture. We approach this question by adapting the well-established mechanistic interpretability techniques of path patching and zero-ablation that allows identifying the effect of different circuits on LLM generations. Our extensive experiments on eight different models and five datasets, representing tasks predominantly requiring factual recall, clearly demonstrate that uncertainty is produced in the same parts of a model that are responsible for the factuality of generated responses. We release code for our implementation.

## 1 Introduction

Uncertainty quantification (UQ) in large language models (LLMs) for knowledge-intensive tasks [16] remains a critical yet understudied area. Despite achieving human-level performance on various benchmarks, LLMs often struggle with reliable uncertainty estimation, leading to issues such as overconfidence and hallucination [19]. This limitation has strong implications for their trustworthiness and safety in high-stakes applications. While recent research has explored verbalized uncertainty in LLMs [1, 10, 11], significant gaps remain in our understanding and ability to improve UQ. In particular, existing UQ techniques typically provide little insight into the factors responsible for an uncertainty estimate, limiting their usefulness both as tools for trustworthiness. We propose leveraging mechanistic interpretability, an approach focused on characterizing models' internal mechanisms of reasoning, to advance our comprehension and enhancement of uncertainty quantification in LLMs.

Following Kadavath et al. [10], to better understand how LMs generate uncertainty estimates, we used parametric $\mathbb{P}(\text{IK})$ ("probability that I know") probes—one-layer binary classifiers that are trained to predict the probability that a given LM knows the answer to a given question. As in [10], we trained $\mathbb{P}(\text{IK})$ probes on several datasets and with several models. We then used these probes' predicted confidences as target metrics for path patching and zero-ablation, two mechanistic interpretability techniques which identify the components of a model relevant for a task by testing the effect of an interventons made on activations in the model during evaluation. We compared the mechanistic signatures of changes in the model's accuracy and the probe's output to evaluate whether the same circuits were responsible for the answer and the predicted confidence.

In our empirical evaluation, in which we performed zero-ablation for a large range of model–dataset combinations and path patching for one combination, we found that model accuracy and probe behavior largely responded to the same interventions, indicating that circuits responsible for the factuality of responses and for uncertainty quantification are located in the same parts of the model.

For a group of knowledge-intensive question answering [16] tasks, model accuracy and probe confidence are (highly) positively related to one another. We conclude that, at least on recall tasks, a

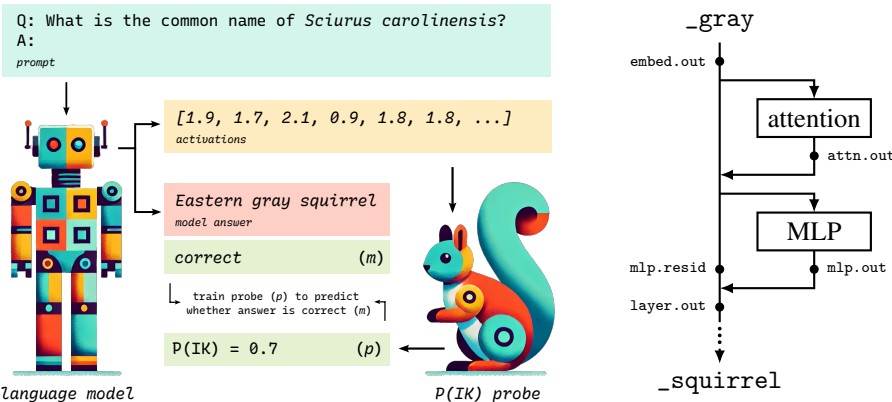

Figure 1: **Left:** $\mathbb{P}$(IK) probing. The LLM takes a question as input and returns an answer and last-layer activations. Answers are checked for correctness. The probe learns to predict whether the model's answer is correct, based on the last-layer activations. Our analysis uses the probe as a proxy for an LLM's $\mathbb{P}$(IK). We conduct path patching and zero ablation studies on the probe and the corresponding LLM. **Right:** Locations used in interventions. Path-patching restorations are at `mlp.resid`, `mlp.out`, `layer.out`, and `embed.out`. Zero-ablations are at `attn.out` and `mlp.out`.

language model's representation of confidence may derive mainly from introspection on its question-answering process, rather than from separate reasoning specific to the UQ task.

To summarize, the key contributions of this paper are as follows:

1. We use mechanistic interpretability and uncertainty quantification tools to investigate the mechanistic sources of uncertainty in large language models. To do so, we use a logistic $\mathbb{P}$(IK) probe with path patching and zero-ablation to perform a hypothesis test to examine whether LLM uncertainty and the factuality of answers generated by an LLM reside in shared or separate circuits within the model.

2. We perform an extensive empirical analysis on eight different models and five recall-intensive datasets, and find evidence that uncertainty quantification and the factuality of answers generated by an LLM are handled by the same parts of the model.

## 2 Background

**Path Patching.** Path patching is a causal intervention method that aims to trace and identify important components in neural models for a given task [14, 18], which is a generalization of causal mediation analysis [17]. In this work, we use path patching [14] to examine the importance and role of individual circuits and components in LLMs. Specifically, given a specific input $q$, path patching involves three runs: (1) a clean run, in which the original input $q$ is given to the model, which is used to obtain the hidden states of each layer; (2) a corrupted run, in which the input embeddings of certain tokens are corrupted by adding noise or (in this paper) replaced with zeros; and (3) a corrupted-with-restoration run, in which the computation is similar to the corrupted run except that the hidden states at specific locations $\ell$ in the model are restored using the hidden states obtained from the clean run. By comparing the differences between the output (predicted probabilities) of the clean, corrupted, and restored runs, path patching allows the identification of important components in LLMs. That is, if the restored run achieves a similar effect as the clean run, it is likely that the corresponding restored component plays an important role in the model's processing.

**Zero-Ablation.** Zero-ablation is a mechanistic intervention technique that takes advantage of a transformer's residual structure by treating attention or MLP layers as separable modules which read from and write to the residual stream [6, 15]. A component $\ell$ (in this paper, an attention or MLP layer) is "ablated" by replacing its output with zero. The drop in model performance on a given task after an intervention removing a component $\ell$ provides a measure of the importance of $\ell$ for the task.

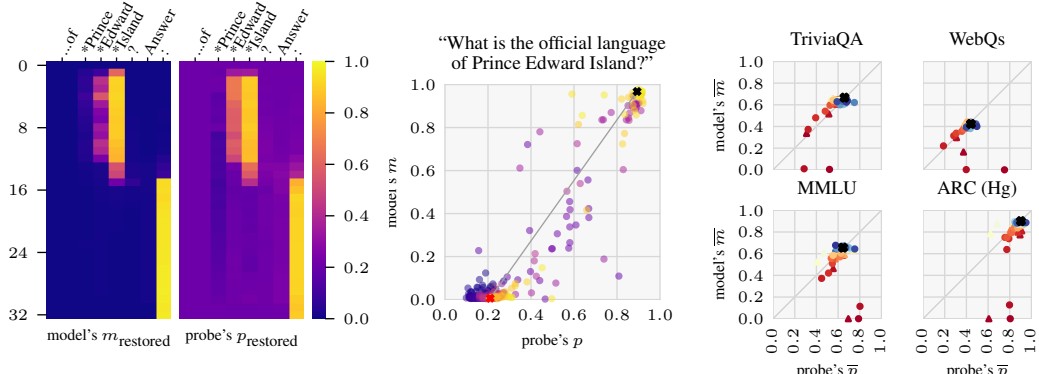

Figure 2: **Left:** Results of path patching for Llama 3 8B Instruct on a question in CounterFact. Only `layer.out` locations are shown (plus `embed.out` in the first row). The input embeddings for the starred tokens are replaced with zeros in the corrupted and restored runs. **Center:** Predicting $m$ given $p$. The black and red X (top-right and bottom-left) show the clean and corrupted runs; all others show restored runs. Yellow points are later in the sequence. The grey line shows the predictor $\hat{m}$. **Right:** Results of zero ablation for Llama 3 8B Instruct on four different datasets. Circle, triangle, and X markers represent MLP ablations, attention ablations, and clean runs respectively. Warmer colors represent earlier layers.

## 3 Uncertainty Introspection: Investigating the Shared Circuits Hypothesis

The aim of this paper is to make progress toward characterizing the mechanistic structures used for UQ in language models. To this end, we propose a theoretical hypothesis ("shared circuits") about the locations of these structures, along with operationalizations which we test experimentally.

> **Shared Circuits Hypothesis.** Uncertainty quantification in question-answering (QA) systems may be carried out in a variety of ways. We hypothesize that language models are capable of expressing uncertainty using **shared circuits** that both solve the underlying question-answering task and output uncertainty information. This contrasts with the possibility that uncertainty quantification emerges in **separate circuits**, either to post-process messy uncertainty signals from question-answering circuits or to do uncertainty calculations of their own.

We study eight Llama[12, 13] and Gemma [7] models and five datasets, described in Appendix **??**.

### 3.1 Experiment Design: Path Patching

On a given question $q_i$ in a dataset $\mathcal{Q}$, for each path patching run (clean, corrupted, and restored) we compute the model's sample probability $m(q_i)$ for the correct first token of the answer, and the probe's confidence $p(q_i)$. (We omit the question for the rest of this section for legibility; we consider questions individually.) Locations $\ell$ where $m_{\text{restored}(\ell)} \approx m_{\text{clean}}$ correspond to parts of the model which are important for solving the QA task; likewise, locations $\ell$ where $p_{\text{restored}(\ell)} \approx p_{\text{clean}}$ correspond to parts of the model which are important for the UQ task.[12] In the context of path patching, we operationalize the shared circuits hypothesis in the claim that $m_{\text{restored}}$ can be predicted from $p_{\text{restored}}$ by interpolating between the clean and corrupted values: for example, if the model's correct-token probability on a restored run is halfway between the values from the clean and corrupted runs, then the probe's confidence should also be halfway between the clean and corrupted runs.

Specifically, for each question $q_i \in \mathcal{Q}$, we claim that the linear predictor $\hat{m}_{\text{restored}}$ defined by

$$\frac{\hat{m}_{\text{restored}(\ell)} - m_{\text{corrupted}}}{m_{\text{clean}} - m_{\text{corrupted}}} = \frac{p_{\text{restored}(\ell)} - p_{\text{corrupted}}}{p_{\text{clean}} - p_{\text{corrupted}}}$$

---

[1] Although note that the converse is not necessarily true; see Appendix D for details.

[2] Here, $\mathbb{P}(\text{corr})_{\text{restored}(\ell)}$ and $\mathbb{P}(\text{IK})_{\text{restored}(\ell)}$ represent the correct token probability and $p$ probe output for a run with the hidden state restored at location $\ell$ in the model; notation is likewise for clean and corrupted runs.

explains most of the variance in $m_{\text{restored}}$ (i.e., has a high coefficient of determination $R^2$). As a (somewhat weak) formalization of this, we attempt to reject the null hypothesis

$$H_0 : R^2 \text{ is no greater than expected under random permutations of the set of locations } \ell. \quad (1)$$

## 3.2 Experiment Design: Zero-Ablation

We also test the shared circuits hypothesis via zero-ablation on layers. Here, because we are interested in multi-token answers, we define $m(q_i)$ as the probability of a correct answer sampled by the model when prompted on the question $q_i \in \mathcal{Q}$, and $p(q_i)$ as the probe output on that question. Taking means over $\mathcal{Q}$, we can compare changes in the model accuracy $\overline{m}$ and the average probe output $\overline{p}$. Under the shared circuits hypothesis, the change in the probe output from ablation $|\overline{p}_{\text{ablated}(\ell)} - p_{\text{clean}}|$ is large when the change in model accuracy $|\overline{m}_{\text{ablated}(\ell)} - m_{\text{clean}}|$ is large. Concretely, we claim that the predictor $\hat{\overline{m}}$ defined by

$$m_{\text{clean}} - \hat{\overline{m}}_{\text{ablated}(\ell)} = |\overline{p}_{\text{ablated}(\ell)} - p_{\text{clean}}|$$

explains most of the variance in $\overline{m}_{\text{ablated}}$ (has a high $R^2$), and attempt to reject the null hypothesis

$$H_0 : R^2 \text{ is no greater than expected under random permutations of the set of layers } \ell. \quad (2)$$

## 3.3 Testing the Hypothesis

**Path Patching.** We performed path patching with Llama 3 8B Instruct [13] on a random sample of 16 questions from the CounterFact dataset [14], considering only questions which the model could answer ($m_{\text{clean}} > 0.5$). We used the probe and few-shot prompt for TriviaQA. Across this sample, the predictors $\hat{m}_{\text{restored}}$ generally estimated $m_{\text{restored}}$ well, with $R^2 > 0.6$ in all but three cases.[3] For each question $q_i$, we tested the null hypothesis (1) by sampling 10,000 permutations.[4] In all cases, we reject $H_0$ with $p < 0.0001$.

**Zero-Ablation.** We performed zero ablation with eight models across five question-answering datasets (see Appendix B). Across this sample, the predictors $\hat{\overline{m}}_{\text{ablated}}$ generally estimated $\overline{m}_{\text{ablated}}$ better than chance, with a median of $R^2 = 0.33$. For each model–dataset combination, we tested the null hypothesis (2) by sampling 10,000 permutations. We reject the null hypothesis with $p < 0.05$ in 36 out of 38 cases, and $p < 0.0001$ in 31 out of 38 cases.

In many cases, the model's uncertainty representation plays particularly nicely with zero-ablation, remaining calibrated on average even after an intervention: using the same statistical framework as above, the very simple predictor $\hat{\overline{m}}_{\text{ablated}} = \overline{p}_{\text{ablated}}$ does better than expected under random permutations in 27 out of 38 cases (at $p < 0.05$).[5] While other explanations may be possible, one interpretation of these results is that a given component makes a nonzero contribution to the model's uncertainty representation if and only if it can also contribute information about the answer.

# 4 Discussion and Conclusion

The results of our path patching and zero-ablation analyses broadly support the shared circuits hypothesis, implying that across the setups we considered the sets of model components used for question-answering and uncertainty quantification were largely, albeit not entirely, the same. This suggests that $\mathbb{P}(\text{IK})$ probing may be a viable way of eliciting introspective, interpretable uncertainty estimates. Based on these findings, further research could analyze the mechanisms responsible for $\mathbb{P}(\text{IK})$ estimates in greater detail or apply $\mathbb{P}(\text{IK})$ probing as an interpretability tool to study phenomena such as hallucination in LLMs and meaningfully contribute to technical AI governance.

---

[3]Based on manual inspection (see graphs in the Supplementary Materials), we conclude that $R^2 < 1$ both due to small discrepancies between UQ and QA circuitry and due to nonlinearity in the UQ/QA relationship.

[4]Specifically, we shuffle the values of $m_{\text{restored}(\ell)}$ independently for the `mlp.out`, `mlp.resid`, and `layer.out/embed.out` locations, to exclude the explanation that the predictor works well because the `mlp.out` and `mlp.resid` states each carry less information than `layer.out`. We do likewise for zero-ablation.

[5]If $R^2$ is the fraction of the variance in $\overline{m}_{\text{ablated}}$ explained by $\hat{\overline{m}}_{\text{ablated}} = \overline{p}_{\text{ablated}}$, we reject the null hypothesis $R^2$ *is no greater than expected under random permutations of the set of layers* $\ell$ at $p < 0.05$ in 27/38 cases.

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

# Appendix

## Appendix A Reproducibility

Code to reproduce our results can be found at

https://anonymous.4open.science/r/sciurus_anonymized-E434/

## Appendix B Models and Datasets

We studied the following eight models and five datasets:

Table 1: Models studied

| Model | Parameters | Layers | Dataset |
|-------|-----------|--------|---------|
| Llama 2 7B | 7B | 32 | TriviaQA[9] |
| Llama 2 7B Chat | 7B | 32 | WebQuestions[2] |
| Llama 2 13B | 13B | 40 | MMLU[8] |
| Llama 2 13B Chat | 13B | 40 | ARC[4] |
| Llama 3 8B | 8B | 32 | CounterFact[14] |
| Llama 3 8B Instruct | 8B | 32 | |
| Gemma 2 2B Instruct | 2B | 26 | |
| Gemma 2 9B Instruct | 9B | 42 | |

All the datasets studied, with the partial exception of MMLU, are "recall-intensive" in that they largely depend on recalling factual information learned during training. Based on some preliminary zero-ablation experiments, we believe that models may exhibit separate circuits on some non-recall tasks such as simple synthetic math questions.

ARC includes both the ARC-Easy and ARC-Challenge splits. ARC questions are drawn from standardized tests; the datasets listed as ARC (Hg) and ARC (Other) correspond, respectively, to the "Mercury" test and to a combination of the other 20 tests.

We reformulated CounterFact prompts as questions to match the format of our other datasets. Because we used the TriviaQA probe for the path patching experiment with CounterFact, we also did few-shot prompting with the prompt from TriviaQA.

### B.1 Licenses

Models:

- Llama 2 is licensed under the Llama 2 Community License Agreement, available at https://ai.meta.com/llama/license/.
- Llama 3 is licensed under the Meta Llama 3 License, available at https://llama.meta.com/llama3/license/.
- Gemma 2 is licensed under the Gemma Terms of Use, available at https://ai.google.dev/gemma/terms.

Datasets:

- TriviaQA is licensed under the Apache License 2.0, available at https://www.apache.org/licenses/LICENSE-2.0.
- WebQuestions is licensed under the Creative Commons Attribution 4.0 International License, available at https://creativecommons.org/licenses/by/4.0/.
- MMLU (Massive Multitask Language Understanding) is licensed under the MIT License, available at https://opensource.org/licenses/MIT.
- ARC (AI2 Reasoning Challenge) is licensed under the Creative Commons Attribution-ShareAlike 4.0 International License, available at https://creativecommons.org/licenses/by-sa/4.0/.
- CounterFact is licensed under the MIT License, available at https://opensource.org/licenses/MIT.

## Appendix C   Probe Design

We use a $\mathbb{P}(\text{IK})$ probing approach in part because of the difficulty of reasoning about uncertainty using token probabilities. Token probabilities for open-ended questions are a highly imperfect proxy for a model's confidence, because they conflate semantic uncertainty (uncertainty about content) with syntactic uncertainty (uncertainty about form). Furthermore, we are most interested in improving uncertainty quantification for fine-tuned chat models, for which token probabilities do not correspond to an underlying distribution over possible text strings.

We construct a dataset on which to train the $\mathbb{P}(\text{IK})$ probe according to the following steps.

1. Perform 32 forward passes for each question on the question-answering task. We used few-shot prompting with 5 examples to ensure that the model answered in the right format.
2. Check whether a model's answers are correct. Specifically, we check whether a model's answer contains any correct answer as a substring, ignoring case.
3. For each question in the dataset, save the number of correct and incorrect answers (implying a "true probability" of the model answering correctly).
4. Also, for each question, save the output of the model's last layer (before the unembedding). This is a vector in $\mathbb{R}^{d_{\text{model}}}$.

The $\mathbb{P}(\text{IK})$ probe is a logistic classifier $p : \mathbb{R}^{d_{\text{model}}} \to (0, 1)$ which takes these last layer activations as input and returns the proportion of correct answers. For example, if the model answers a question correctly 47% of the time, the probe should output 0.47 when given the last-layer activations at the last token of that question. We trained with binary cross-entropy loss, using dropout and a triangular learning rate schedule, and used a low learning rate ($\eta = 3 \times 10^{-6}$) as in [10].

## Appendix D   Limitations of Path Patching and Zero-Ablation

Path patching and zero-ablation, like many interpretability techniques, yield results which can imperfectly reflect the contributions of model internals to a task. In particular:

**Zero-ablation.** We chose to ablate activations in the model with zeros. While the zero vector is far from an arbitrary choice, especially given its relevance to dropout and the additive residual structure of a transformer, this approach may lack specificity. For example, zero-ablating an early or late MLP layer sometimes severely damages a model's ability to produce coherent language in general, so accuracies from ablation do not necessarily correspond to the flow of question-specific information through the model. Approaches such as causal scrubbing [3] avoid this limitation but are generally more computationally expensive.

**Path patching.** The "path" through the model identified comprises, to a first approximation, the set of points in the model at which *all* information relevant to the task is present. As such, when information relevant to a question passes along multiple paths in parallel, it may be that no individual path shows a substantial difference between the restored and baseline conditions. For example, in the question in Fig. 2 (left), restoring the input embedding for any one token of "Prince Edward Island" without the others has little effect on the model.

 # Appendix E    Full Results for Zero-Ablation

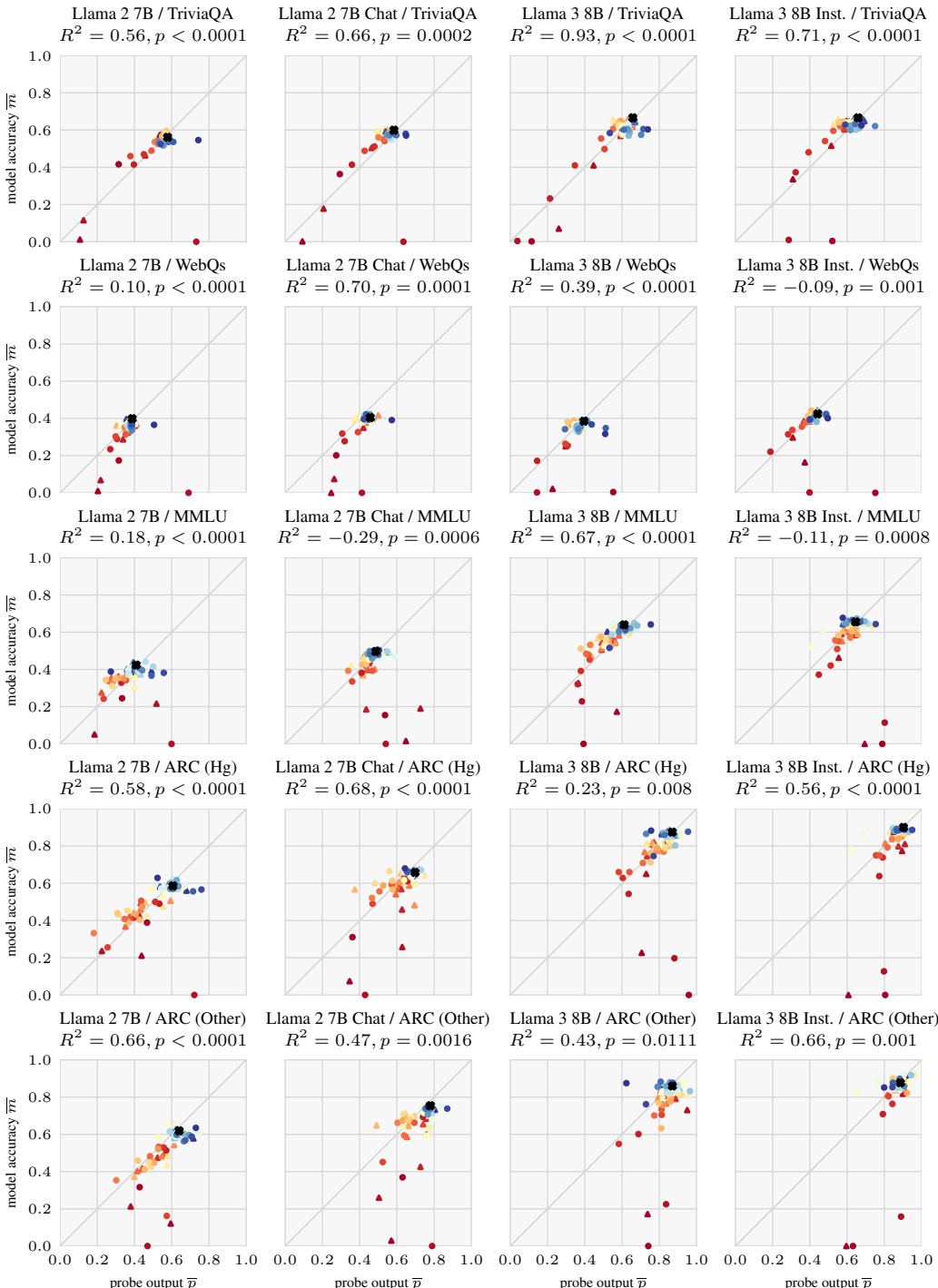

Figure 3:  Results of zero-ablation for eight models and five datasets. Circle, triangle, and X markers represent MLP ablations, attention ablations, and clean runs respectively. Warmer colors represent earlier layers. Error bars for individual points are omitted for legibility, but std. err. $< 0.032$ in all cases (by the bounds on $p$ and $m$).

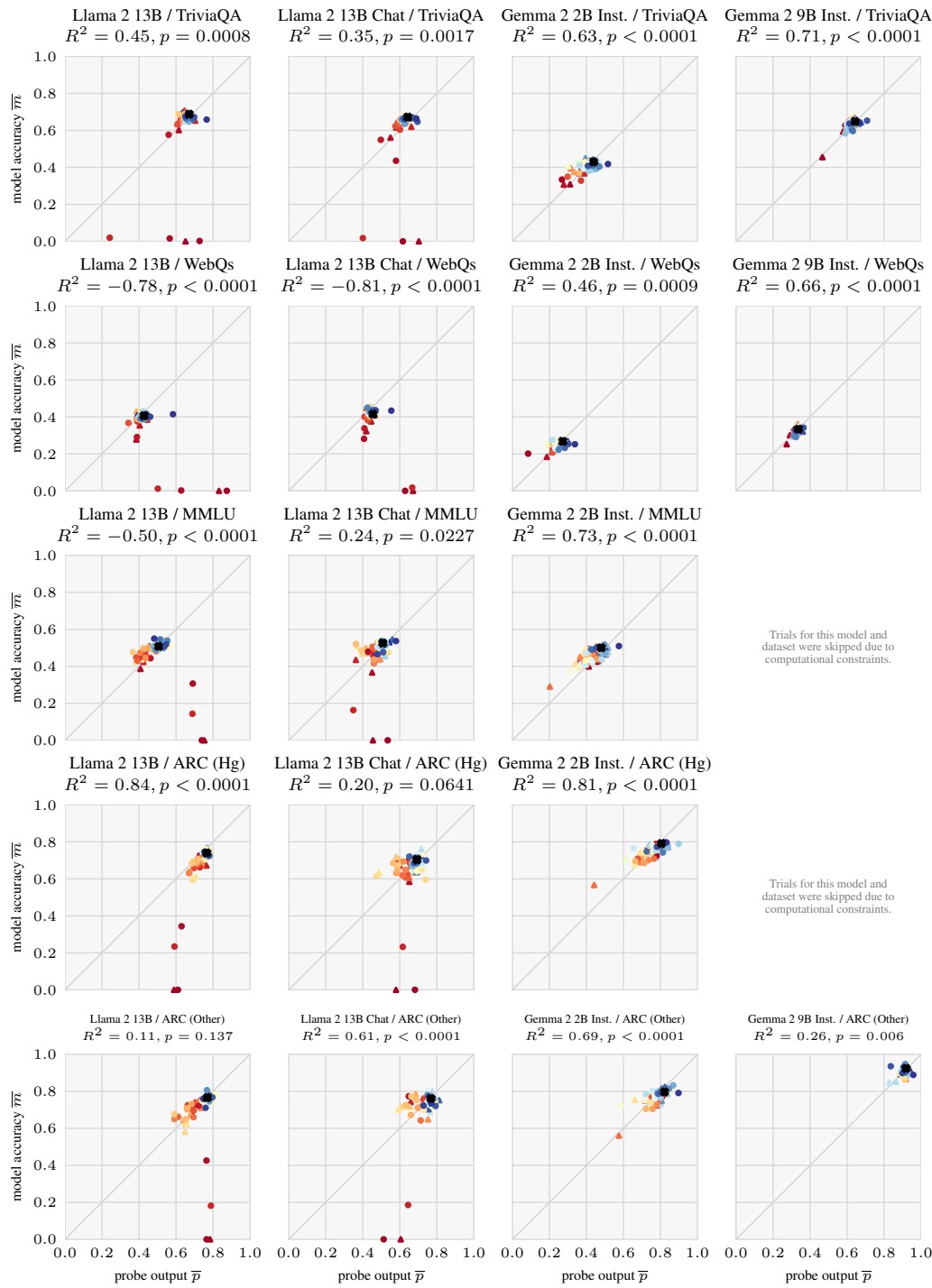

Figure 4: *(continued)* Results of zero-ablation for eight models and five datasets. Circle, triangle, and X markers represent MLP ablations, attention ablations, and clean runs respectively. Warmer colors represent earlier layers. Error bars for individual points are omitted for legibility, but std. err. $< 0.032$ in all cases (by the bounds on $p$ and $m$).

## Appendix F    Computational Resources

This project has used approximately 1000 GPU-hours of computation time on an academic cluster, mainly on RTX8000 GPUs with 48 GB of memory, including approximately 500 GPU-hours for results used directly in this paper. Results for individual model/dataset combinations can be reproduced independently; for example, the code to produce the TriviaQA / Llama 3 8B Instruct results ran in approximately 20 GPU-hours.

## Appendix G    Ethics Statement

This paper intends to advance the areas of interpretability and uncertainty quantification for language models, with the primary aim of making language models more reliable and more trustworthy. We expect these research directions in general to reduce societal risks from machine learning (for example, by allowing for warning signals in situations where a model might be lying or making a dangerous mistake). Nevertheless, since reliability work also makes systems more useful, some caution is warranted: for example, users might be tempted to deploy the resultant more-reliable systems in higher-stakes contexts in which tail risks from failures are greater.

The humanoid and sciuroid robots in Fig. 1 were created using DALL-E 3.

