# OpenReview forum: "SCIURus: Shared Circuits for Interpretable Uncertainty Representations in Language Models"
_NeurIPS.cc/2024/Workshop/SafeGenAi — SafeGenAi Poster_

### Official Review · Reviewer_Xht6 · 2024-10-08
**The paper tackles an important problem in large language models (LLMs), but the method and results are unclear, not well-structured, and lack some essential details; it also requires significant improvement in writing.**

**Rating:** 4
**Confidence:** 5

**Review:**

**Pros**:

- The paper addresses a critical area in LLM research, focusing on the mechanistic sources of uncertainty.
- The proposed method is straightforward to implement and computationally inexpensive, making it feasible for testing new models efficiently.

**Cons**:

- The explanation of the method and experiments lacks clarity, particularly the description of how path-patching and zero-ablation are conducted.
- Although the paper claims to experiment on 5 models and 8 datasets, only Llama 3 8B with one dataset is detailed in the main text.
- Figures are not properly referenced in the text, such as Figures 1 and 2.
- The approach requires access to the final activation layer of the LLM, which is not available in widely used models like ChatGPT.
- The paper would be more valuable if it proposed a method to improve uncertainty quantification in LLMs, beyond just analyzing it.

---

### Official Review · Reviewer_Syv1 · 2024-10-09
**Clear and crisp contribution towards understanding uncertainty mechanisms**

**Rating:** 8
**Confidence:** 3

**Review:**

The paper is well-written and makes a clear contribution to understanding mechanisms of uncertainty in language models.  The hypothesis, methods, and results are all well-defined and connected.  The results in the paper provide a deeper understanding of uncertainty, which is important in model safety and are significant and valuable for this workshop.  Extending the paper to additional models (and/or additional datasets) would help further strengthen its results.

Strengths:

* The paper is well-written and succinct, and makes a clear contribution to understanding the relationship between uncertainty estimation paths and QA paths
* The logic between the hypothesis, methodology, and results is very clear.  The authors describe how each experiment informs answering the hypothesis.
* The results clearly support the conclusion, and include statistics that communicate the significance of results.


Clarifications/Weaknesses:

* Line 29: “interventons” → “interventions”
* Generally the methodology was clear and easy to understand.  However, I did need to read the Appendix to fully understand the methods.  Given that there is room in the paper, moving some of this content to the main body might help readers.
* Given the compelling methodology and results, I would be interested to see how this applies to other datasets and/or models.

---

### Official Review · Reviewer_Q3FN · 2024-10-09
**Review for the Paper: "SCIURus: Shared Circuits for Interpretable Uncertainty Representations in Language Models"**

**Rating:** 5
**Confidence:** 3

**Review:**

### **Summary**

The paper investigates the relationship between uncertainty quantification (UQ) and factuality in large language models (LLMs). It introduces the "shared circuits" hypothesis, suggesting that the same model circuits responsible for answering factual questions are also responsible for expressing uncertainty. Using mechanistic interpretability techniques, such as path patching and zero-ablation, the authors analyze the internal components of eight different models across five datasets to validate this hypothesis. Their results indicate that uncertainty and factuality tend to be processed by overlapping components in the model, suggesting that introspective uncertainty is derived from the same circuits as the answer-generation process.

### **Strengths**

1. **Innovative Use of Mechanistic Interpretability**: The authors utilize cutting-edge techniques like path patching and zero-ablation to probe into model behavior. These methods provide a deeper understanding of the model's internal decision-making processes, going beyond superficial output analysis
2. **Clear Hypothesis Testing**: The paper establishes a well-defined hypothesis—the shared circuits hypothesis—and rigorously tests it using statistical methods. This clarity of purpose strengthens the scientific merit of the work

### **Weaknesses**

1. **Limited Task Scope**: The experiments focus on factual recall tasks, which may limit the generalizability of the shared circuits hypothesis. It remains unclear whether this finding holds for other types of tasks, such as reasoning or creative text generation
2. **Interpretability Challenges**: While the mechanistic techniques used are powerful, they can be computationally expensive and may not be easily accessible to a wider audience. The results, though significant, might not translate easily into practical, everyday applications of LLMs without further simplification
3. **Ablation Methods and Specificity**: Zero-ablation, while informative, can sometimes introduce artifacts by severely disrupting a model's performance, making it harder to isolate the exact components responsible for uncertainty. The paper acknowledges this limitation, but it could have explored alternative ablation techniques to improve result clarity

### **Detailed Comments**

- The authors are transparent about the limitations of their methods, particularly the potential noisiness of zero-ablation and the computational cost of mechanistic interpretability techniques. Future work could benefit from exploring tasks that go beyond factual recall to test whether the shared circuits hypothesis holds in different contexts. Furthermore, the paper could explore more lightweight methods of probing model uncertainty, as the current approach might be impractical for real-time applications
- The paper is generally well-written, with a logical structure and clear presentation of methods and results. However, some sections, particularly those on path patching, could benefit from additional explanations for readers unfamiliar with advanced interpretability techniques. The graphs and tables, while informative, might be made more accessible with more detailed legends and explanations